# EDML for Learning Parameters in Directed and Undirected Graphical Models

**Khaled S. Refaat, Arthur Choi, Adnan Darwiche**
Computer Science Department
University of California, Los Angeles
{krefaat,aychoi,darwiche}@cs.ucla.edu

## Abstract

EDML is a recently proposed algorithm for learning parameters in Bayesian networks. It was originally derived in terms of approximate inference on a meta-network, which underlies the Bayesian approach to parameter estimation. While this initial derivation helped discover EDML in the first place and provided a concrete context for identifying some of its properties (e.g., in contrast to EM), the formal setting was somewhat tedious in the number of concepts it drew on. In this paper, we propose a greatly simplified perspective on EDML, which casts it as a general approach to continuous optimization. The new perspective has several advantages. First, it makes immediate some results that were non-trivial to prove initially. Second, it facilitates the design of EDML algorithms for new graphical models, leading to a new algorithm for learning parameters in Markov networks. We derive this algorithm in this paper, and show, empirically, that it can sometimes learn estimates more efficiently from complete data, compared to commonly used optimization methods, such as conjugate gradient and L-BFGS.

## 1   Introduction

EDML is a recently proposed algorithm for learning MAP parameters of a Bayesian network from incomplete data [5, 16]. While it is procedurally very similar to Expectation Maximization (EM) [7, 11], EDML was shown to have certain advantages, both theoretically and practically. Theoretically, EDML can in certain specialized cases provably converge in one iteration, whereas EM may require many iterations to solve the same learning problem. Some empirical evaluations further suggested that EDML and hybrid EDML/EM algorithms can sometimes find better parameter estimates than vanilla EM, in fewer iterations and less time. EDML was originally derived in terms of approximate inference on a meta-network used for Bayesian approaches to parameter estimation. This graphical representation of the estimation problem lent itself to the initial derivation of EDML, as well to the identification of certain key theoretical properties, such as the one we just described. The formal details, however, can be somewhat tedious as EDML draws on a number of different concepts. We review EDML in such terms in the supplementary appendix.

In this paper, we propose a new perspective on EDML, which views it more abstractly in terms of a simple method for continuous optimization. This new perspective has a number of advantages. First, it makes immediate some results that were previously obtained for EDML, but through some effort. Second, it facilitates the design of new EDML algorithms for new classes of models, where graphical formulations of parameter estimation, such as meta-networks, are lacking. Here, we derive, in particular, a new parameter estimation algorithm for Markov networks, which is in many ways a more challenging task, compared to the case of Bayesian networks. Empirically, we find that EDML is capable of learning parameter estimates, under complete data, more efficiently than popular methods such as conjugate-gradient and L-BFGS, and in some cases, by an order-of-magnitude.

This paper is structured as follows. In Section 2, we highlight a simple iterative method for, approximately, solving continuous optimization problems. In Section 3, we formulate the EDML algorithm for parameter estimation in Bayesian networks, as an instance of this optimization method. In Section 4, we derive a new EDML algorithm for Markov networks, based on the same perspective. In Section 5, we contrast the two EDML algorithms for directed and undirected graphical models, in the complete data case. We empirically evaluate our new algorithm for parameter estimation under complete data in Markov networks, in Section 6; review related work in Section 7; and conclude in Section 8. Proofs of theorems appear in the supplementary appendix.

## 2 An Approximate Optimization of Real-Valued Functions

Consider a real-valued objective function $f(x)$ whose input $x$ is a vector of components:

$$x = (x_1, \ldots, x_i, \ldots, x_n),$$

where each component $x_i$ is a vector in $\mathbb{R}^{k_i}$ for some $k_i$. Suppose further that we have a constraint on the domain of function $f(x)$ with a corresponding function $g$ that maps an arbitrary point $x$ to a point $g(x)$ satisfying the given constraint. We say in this case that $g(x)$ is a *feasibility function* and refer to the points in its range as *feasible points.*

Our goal here is to find a feasible input vector $x = (x_1, \ldots, x_i, \ldots, x_n)$ that optimizes the function $f(x)$. Given the difficulty of this optimization problem in general, we will settle for finding stationary points $x$ in the constrained domain of function $f(x)$.

One approach for finding such stationary points is as follows. Let $x^\star = (x_1^\star, \ldots, x_i^\star, \ldots, x_n^\star)$ be a feasible point in the domain of function $f(x)$. For each component $x_i$, we define a sub-function

$$f_{x^\star}(x_i) = f(x_1^\star, \ldots, x_{i-1}^\star, x_i, x_{i+1}^\star, \ldots, x_n^\star).$$

That is, we use the $n$-ary function $f(x)$ to generate $n$ sub-functions $f_{x^\star}(x_i)$. Each of these sub-functions is obtained by fixing all inputs $x_j$ of $f(x)$, for $j \neq i$, to their values in $x^\star$, while keeping the input $x_i$ free. We further assume that these sub-functions are subject to the same constraints that the function $f(x)$ is subject to.

We can now characterize all feasible points $x^\star$ that are stationary with respect to the function $f(x)$, in terms of local conditions on sub-functions $f_{x^\star}(x_i)$.

**Claim 1** *A feasible point $x^\star = (x_1^\star, \ldots, x_i^\star, \ldots, x_n^\star)$ is stationary for function $f(x)$ iff for all $i$, component $x_i^\star$ is stationary for sub-function $f_{x^\star}(x_i)$.*

This is immediate from the definition of a stationary point. Assuming no constraints, at a stationary point $x^\star$, the gradient $\nabla f(x^\star) = 0$, i.e., $\nabla_{x_i} f(x^\star) = \nabla f_{x^\star}(x_i^\star) = 0$ for all $x_i$, where $\nabla_{x_i} f(x^\star)$ denotes the sub-vector of gradient $\nabla f(x^\star)$ with respect to component $x_i$.[1]

With these observations, we can now search for feasible stationary points $x^\star$ of the constrained function $f(x)$ using an iterative method that searches instead for stationary points of the constrained sub-functions $f_{x^\star}(x_i)$. The method works as follows:

1. Start with some feasible point $x^t$ of function $f(x)$ for $t = 0$
2. While some $x_i^t$ is not a stationary point for constrained sub-function $f_{x^t}(x_i)$
   (a) Find a stationary point $y_i^{t+1}$ for each constrained sub-function $f_{x^t}(x_i)$
   (b) $x^{t+1} = g(y^{t+1})$
   (c) Increment $t$

The real computational work of this iterative procedure is in Steps 2(a) and 2(b), although we shall see later that such steps can, in some cases, be performed efficiently. With an appropriate feasibility function $g(y)$, one can guarantee that a fixed-point of this procedure yields a stationary point of the constrained function $f(x)$, by Claim 1.[2] Further, any stationary point is trivially a fixed-point of this procedure (one can seed this procedure with such a point).

As we shall show in the next section, the EDML algorithm—which has been proposed for parameter estimation in Bayesian networks—is an instance of the above procedure with some notable observations: (1) the sub-functions $f_{x^t}(x_i)$ are convex and have unique optima; (2) these sub-functions have an interesting semantics, as they correspond to posterior distributions that are induced by Naive Bayes networks with soft evidence asserted on them; (3) defining these sub-functions requires inference in a Bayesian network parameterized by the current feasible point $x^t$; (4) there are already several convergent, fixed-point iterative methods for finding the unique optimum of these sub-functions; and (5) these convergent methods produce solutions that are always feasible and, hence, the feasibility function $g(y)$ corresponds to the identity function $g(y) = y$ in this case.

We next show this connection to EDML as proposed for parameter estimation in Bayesian networks. We follow by deriving an EDML algorithm (another instance of the above procedure), but for parameter estimation in undirected graphical models. We will also study the impact of having complete data on both versions of the EDML algorithm, and finally evaluate the new instance of EDML by comparing it to conjugate gradient and L-BFGS when applied to complete datasets.

## 3 EDML for Bayesian Networks

From here on, we use upper case letters $(X)$ to denote variables and lower case letters $(x)$ to denote their values. Variable sets are denoted by bold-face upper case letters $(\mathbf{X})$ and their instantiations by bold-face lower case letters $(\mathbf{x})$. Generally, we will use $X$ to denote a variable in a Bayesian network and $\mathbf{U}$ to denote its parents. A network parameter will therefore have the general form $\theta_{x|\mathbf{u}}$, representing the probability $Pr(X\!=\!x|\mathbf{U}\!=\!\mathbf{u})$.

Consider a (possibly incomplete) dataset $\mathcal{D}$ with examples $\mathbf{d}_1, \ldots, \mathbf{d}_N$, and a Bayesian network with parameters $\theta$. Our goal is to find parameter estimates $\theta$ that minimize the negative log-likelihood:

$$f(\theta) = -\ell\ell(\theta|\mathcal{D}) = -\sum_{i=1}^{N} \log Pr_\theta(\mathbf{d}_i). \tag{1}$$

Here, $\theta = (\ldots, \theta_{X|\mathbf{u}}, \ldots)$ is a vector over the network parameters. Moreover, $Pr_\theta$ is the distribution induced by the Bayesian network structure under parameters $\theta$. As such, $Pr_\theta(\mathbf{d}_i)$ is the probability of observing example $\mathbf{d}_i$ in dataset $\mathcal{D}$ under parameters $\theta$.

Each component of $\theta$ is a parameter set $\theta_{X|\mathbf{u}}$, which defines a parameter $\theta_{x|\mathbf{u}}$ for each value $x$ of variable $X$ and instantiation $\mathbf{u}$ of its parents $\mathbf{U}$. The feasibility constraint here is that each component $\theta_{X|\mathbf{u}}$ satisfies the convex sum-to-one constraint: $\sum_x \theta_{x|\mathbf{u}} = 1$.

The above parameter estimation problem is clearly in the form of the constrained optimization problem that we phrased in the previous section and, hence, admits the same iterative procedure proposed in that section for finding stationary points. The relevant questions now are: What form do the sub-functions $f_{\theta^\star}(\theta_{X|\mathbf{u}})$ take in this context? What are their semantics? What properties do they have? How do we find their stationary points? What is the feasibility function $g(y)$ in this case? Finally, what is the connection to previous work on EDML? We address these questions next.

### 3.1 Form

We start by characterizing the sub-functions of the negative log-likelihood given in Equation 1.

**Theorem 1** *For each parameter set $\theta_{X|\mathbf{u}}$, the negative log-likelihood of Equation 1 has the sub-function:*

$$f_{\theta^\star}(\theta_{X|\mathbf{u}}) = -\sum_{i=1}^{N} \log \left( C_{\mathbf{u}}^i + \sum_x C_{x|\mathbf{u}}^i \cdot \theta_{x|\mathbf{u}} \right) \tag{2}$$

*where $C_{\mathbf{u}}^i$ and $C_{x|\mathbf{u}}^i$ are constants that are independent of parameter set $\theta_{X|\mathbf{u}}$, given by*

$$C_{\mathbf{u}}^i = Pr_{\theta^\star}(\mathbf{d}_i) - Pr_{\theta^\star}(\mathbf{u}, \mathbf{d}_i) \qquad and \qquad C_{x|\mathbf{u}}^i = Pr_{\theta^\star}(x, \mathbf{u}, \mathbf{d}_i)/\theta_{x|\mathbf{u}}^\star$$

To compute the constants $C^i$, we *require inference* on a Bayesian network with parameters $\theta^\star$.[3]

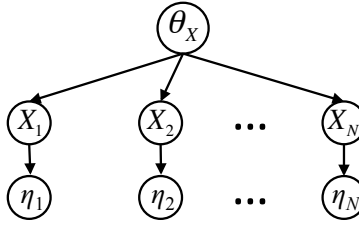

Figure 1: Estimation given independent soft observations.

## 3.2 Semantics

Equation 2 has an interesting semantics, as it corresponds to the negative log-likelihood of a root variable in a naive Bayes structure, on which soft, not necessarily hard, evidence is asserted [5].[4]

This model is illustrated in Figure 1, where our goal is to estimate a parameter set $\theta_X$, given soft observations $\eta = (\eta_1, \ldots, \eta_N)$ on variables $X_1, \ldots, X_N$, where each $\eta_i$ has a strength specified by a weight on each value $x_i$ of $X_i$. If we denote the distribution of this model by $\mathbb{P}$, then (1) $\mathbb{P}(\theta)$ denotes a prior over parameters sets,[5] (2) $\mathbb{P}(x_i|\theta_X = (\ldots, \theta_x, \ldots)) = \theta_x$, and (3) weights $\mathbb{P}(\eta_i|x_i)$ denote the strengths of soft evidence $\eta_i$ on value $x_i$. The log likelihood of our soft observations $\eta$ is:

$$\log \mathbb{P}(\eta|\theta_X) = \sum_{i=1}^{N} \log \sum_{x_i} \mathbb{P}(\eta_i|x_i)\mathbb{P}(x_i|\theta_X) = \sum_{i=1}^{N} \log \sum_{x_i} \mathbb{P}(\eta_i|x_i) \cdot \theta_x \qquad (3)$$

The following result connects Equation 2 to the above likelihood of a soft dataset, when we now want to estimate the parameter set $\theta_{X|\mathbf{u}}$, for a particular variable $X$ and parent instantiation $\mathbf{u}$.

**Theorem 2** *Consider Equations 2 and 3, and assume that each soft evidence $\eta_i$ has the strength $\mathbb{P}(\eta_i|x_i) = C_{\mathbf{u}}^i + C_{x|\mathbf{u}}^i$. It then follows that*

$$f_{\theta^\star}(\theta_{X|\mathbf{u}}) = -\log \mathbb{P}(\eta|\theta_{X|\mathbf{u}}) \qquad (4)$$

This theorem yields the following interesting semantics for EDML sub-functions. Consider a parameter set $\theta_{X|\mathbf{u}}$ and example $\mathbf{d}_i$ in our dataset. The example can then be viewed as providing "votes" on what this parameter set should be. In particular, the vote of example $\mathbf{d}_i$ for value $x$ takes the form of a soft evidence $\eta_i$ whose strength is given by

$$\mathbb{P}(\eta_i|x_i) = Pr_{\theta^\star}(\mathbf{d}_i) - Pr_{\theta^\star}(\mathbf{u}, \mathbf{d}_i) + Pr_{\theta^\star}(x, \mathbf{u}, \mathbf{d}_i)/\theta_{x|\mathbf{u}}^\star$$

The sub-function is then aggregating these votes from different examples and producing a corresponding objective function on parameter set $\theta_{X|\mathbf{u}}$. EDML optimizes this objective function to produce the next estimate for each parameter set $\theta_{X|\mathbf{u}}$.

## 3.3 Properties

Equation 2 is a convex function, and thus has a unique optimum.[6] In particular, we have logs of a linear function, which are each concave. The sum of two concave functions is also concave, thus our sub-function $f_{\theta^\star}(\theta_{X|\mathbf{u}})$ is convex, and is subject to a convex sum-to-one constraint [16]. Convex functions are relatively well understood, and there are a variety of methods and systems that can be used to optimize Equation 2; see, e.g., [3]. We describe one such approach, next.

## 3.4 Finding the Unique Optima

In every EDML iteration, and for each parameter set $\theta_{X|\mathbf{u}}$, we seek the unique optimum for each sub-function $f_{\theta^\star}(\theta_{X|\mathbf{u}})$, given by Equation 2. Refaat, et al., has previously proposed a fixed-point

algorithm that monotonically improves the objective, and is guaranteed to converge [16]. Moreover, the solutions it produces already satisfy the convex sum-to-one constraint and, hence, the feasibility function $g$ ends up being the identity function $g(\theta) = \theta$.

In particular, we start with some initial feasible estimates $\theta^t_{X|\mathbf{u}}$ at iteration $t = 0$, and then apply the following update equation until convergence:

$$\theta^{t+1}_{x|\mathbf{u}} = \frac{1}{N} \sum_{i=1}^{N} \frac{(C^i_{\mathbf{u}} + C^i_{x|\mathbf{u}}) \cdot \theta^t_{x|\mathbf{u}}}{C^i_{\mathbf{u}} + \sum_{x'} C^i_{x'|\mathbf{u}} \cdot \theta^t_{x'|\mathbf{u}}} \tag{5}$$

Note here that constants $C^i$ are *computed by inference* on a Bayesian network structure under parameters $\theta^t$ (see Theorem 1 for the definitions of these constants). Moreover, while the above procedure is convergent when optimizing sub-functions $f_{\theta^\star}(\theta_{X|\mathbf{u}})$, the global EDML algorithm that is optimizing function $f(\theta)$ may not be convergent in general.

## 3.5 Connection to Previous Work

EDML was originally derived by applying an approximate inference algorithm to a meta-network, which is typically used in Bayesian approaches to parameter estimation [5, 16]. This previous formulation of EDML, which is specific to Bayesian networks, now falls as a special instance of the one given in Section 2. In particular, the "sub-problems" defined by the original EDML [5, 16] correspond precisely to the sub-functions $f_{\theta^\star}(\theta_{X|\mathbf{u}})$ described here. Further, both versions of EDML are procedurally identical when they both use the same method for optimizing these sub-functions.

The new formulation of EDML is more transparent, however, at least in revealing certain properties of the algorithm. For example, it now follows immediately (from Section 2) that the fixed points of EDML are stationary points of the log-likelihood—a fact that was not proven until [16], using a technique that appealed to the relationship between EDML and EM. Moreover, the proof that EDML under complete data will converge immediately to the optimal estimates is also now immediate (see Section 5). More importantly though, this new formulation provides a systematic procedure for deriving new instances of EDML for additional models, beyond Bayesian networks. Indeed, in the next section, we use this procedure to derive an EDML instance for Markov networks, which is followed by an empirical evaluation of the new algorithm under complete data.

## 4   EDML for Undirected Models

In this section, we show how parameter estimation for undirected graphical models, such as Markov networks, can also be posed as an optimization problem, as described in Section 2.

For Markov networks, $\theta = (\ldots, \theta_{\mathbf{X}_a}, \ldots)$ is a vector over the network parameters. Component $\theta_{\mathbf{X}_a}$ is a parameter set for a (tabular) factor $a$, assigning a number $\theta_{\mathbf{x}_a} \geq 0$ for each instantiation $\mathbf{x}_a$ of variables $\mathbf{X}_a$. The negative log-likelihood $-\ell\ell(\theta|\mathcal{D})$ for a Markov network is:

$$-\ell\ell(\theta|\mathcal{D}) = N \log Z_\theta - \sum_{i=1}^{N} \log Z_\theta(\mathbf{d}_i) \tag{6}$$

where $Z_\theta$ is the partition function, and where $Z_\theta(\mathbf{d}_i)$ is the partition function after conditioning on example $\mathbf{d}_i$, under parameterization $\theta$. Sub-functions with respect to Equation 6 may not be convex, as was the case in Bayesian networks. Consider instead the following objective function, which we shall subsequently relate to the negative log-likelihood:

$$f(\theta) = -\sum_{i=1}^{N} \log Z_\theta(\mathbf{d}_i), \tag{7}$$

with a feasibility constraint that the partition function $Z_\theta$ equals some constant $\alpha$. The following result tells us that it suffices to optimize Equation 7 under the given constraint, to optimize Equation 6.

**Theorem 3** *Let $\alpha$ be a positive constant, and let $g(\theta)$ be a (feasibility) function satisfying $Z_{g(\theta)} = \alpha$ and $g(\theta_{\mathbf{x}_a}) \propto \theta_{\mathbf{x}_a}$ for all $\theta_{\mathbf{x}_a}$.[7] For every point $\theta$, if $g(\theta)$ is optimal for Equation 7, subject to its*

*constraint, then it is also optimal for Equation 6. Moreover, a point $\theta$ is stationary for Equation 6 iff the point $g(\theta)$ is stationary for Equation 7, subject to its constraint.*

With Equation 7 as a new (constrained) objective function for estimating the parameters of a Markov network, we can now cast it in the terms of Section 2. We start by characterizing its sub-functions.

**Theorem 4** *For a given parameter set $\theta_{\mathbf{X}_a}$, the objective function of Equation 7 has sub-functions:*

$$f_{\theta^\star}(\theta_{\mathbf{X}_a}) = -\sum_{i=1}^{N} \log \sum_{\mathbf{x}_a} C_{\mathbf{x}_a}^i \cdot \theta_{\mathbf{x}_a} \qquad \text{subject to} \qquad \sum_{\mathbf{x}_a} C_{\mathbf{x}_a} \cdot \theta_{\mathbf{x}_a} = \alpha \qquad (8)$$

*where $C_{\mathbf{x}_a}^i$ and $C_{\mathbf{x}_a}$ are constants that are independent of the parameter set $\theta_{\mathbf{X}_a}$:*

$$C_{\mathbf{x}_a}^i = Z_{\theta^\star}(\mathbf{x}_a, \mathbf{d}_i)/\theta_{\mathbf{x}_a}^\star \qquad and \qquad C_{\mathbf{x}_a} = Z_{\theta^\star}(\mathbf{x}_a)/\theta_{\mathbf{x}_a}^\star.$$

Note that computing these constants *requires inference* on a Markov network with parameters $\theta^\star$.[8]

Interestingly, this sub-function is convex, as well as the constraint (which is now linear), resulting in a unique optimum, as in Bayesian networks. However, even when $\theta^\star$ is a feasible point, the unique optima of these sub-functions may not be feasible when combined. Thus, the feasibility function $g(\theta)$ of Theorem 3 must be utilized in this case.

We now have another instance of the iterative algorithm proposed in Section 2, but for undirected graphical models. That is, *we have just derived an EDML algorithm for such models.*

## 5 EDML under Complete Data

We consider now how EDML simplifies under complete data for both Bayesian and Markov networks, identifying forms and properties of the corresponding sub-functions under complete data.

We start with Bayesian networks. Consider a variable $X$, and a parent instantiation $\mathbf{u}$; and let $\mathcal{D}\#(x\mathbf{u})$ represent the number of examples that contain $x\mathbf{u}$ in the complete dataset $\mathcal{D}$. Equation 2 of Theorem 1 then reduces to: $f_{\theta^\star}(\theta_{X|\mathbf{u}}) = -\sum_x \mathcal{D}\#(x\mathbf{u}) \log \theta_{x|\mathbf{u}} + C$, where $C$ is a constant that is independent of parameter set $\theta_{X|\mathbf{u}}$. Assuming that $\theta^\star$ is feasible (i.e., each $\theta_{X|\mathbf{u}}$ satisfies the sum-to-one constraint), the unique optimum of this sub-function is $\theta_{x|\mathbf{u}} = \frac{\mathcal{D}\#(x\mathbf{u})}{\mathcal{D}\#(\mathbf{u})}$, which is guaranteed to yield a feasible point $\theta$, globally. Hence, EDML produces the unique optimal estimates in its first iteration and terminates immediately thereafter.

The situation is different, however, for Markov networks. Under a complete dataset $\mathcal{D}$, Equation 8 of Theorem 4 reduces to: $f_{\theta^\star}(\theta_{\mathbf{X}_a}) = -\sum_{\mathbf{x}_a} \mathcal{D}\#(\mathbf{x}_a) \log \theta_{\mathbf{x}_a} + C$, where $C$ is a constant that is independent of parameter set $\theta_{\mathbf{X}_a}$. Assuming that $\theta^\star$ is feasible (i.e., satisfies $Z_{\theta^\star} = \alpha$), the unique optimum of this sub-function has the closed form: $\theta_{\mathbf{x}_a} = \frac{\alpha}{N} \frac{\mathcal{D}\#(\mathbf{x}_a)}{C_{\mathbf{x}_a}}$, which is equivalent to the unique optimum one would obtain in a sub-function for Equation 6 [15, 13]. Contrary to Bayesian networks, the collection of these optima for different parameter sets do not necessarily yield a feasible point $\theta$. Hence, the feasibility function $g$ of Theorem 3 must be applied here. The resulting feasible point, however, may no longer be a stationary point for the corresponding sub-functions, leading EDML to iterate further. Hence, under complete data, EDML for Bayesian networks converges immediately, while EDML for Markov networks may take multiple iterations.

Both results are consistent with what is already known in the literature on parameter estimation for Bayesian and Markov networks. The result on Bayesian networks is useful in confirming that EDML performs optimally in this case. *The result for Markov networks, however, gives rise to a new algorithm for parameter estimation under complete data.* We evaluate the performance of this new EDML algorithm after considering the following example.

Let $\mathcal{D}$ be a complete dataset over three variables $A$, $B$ and $C$, specified in terms of the number of times that each instantiation $a, b, c$ appears in $\mathcal{D}$. In particular, we have the following counts:

---

normalize each parameter set to sum-to-one, but then update the constant $\alpha = Z_{\theta^t}$ for the subsequent iteration.

[8]Theorem 4 assumes that $\theta_{\mathbf{x}_a}^\star \neq 0$. In general, $C_{\mathbf{x}_a}^i = \frac{\partial Z_{\theta^\star}(\mathbf{d}_i)}{\partial \theta_{\mathbf{x}_a}}$, and $C_{\mathbf{x}_a} = \frac{\partial Z_{\theta^\star}}{\partial \theta_{\mathbf{x}_a}}$. See also Footnote 3.

Table 1: Speed-up results of EDML over CG and L-BFGS

| problem | #vars | $i_{\text{cg}}$ | $i_{\text{edml}}$ | $t_{\text{cg}}$ | $(S)$ | $i_{\text{l-bfgs}}$ | $i'_{\text{edml}}$ | $t_{\text{l-bfgs}}$ | $(S')$ |
|---|---|---|---|---|---|---|---|---|---|
| zero | 256 | 45 | 105 | 3.62 | 3.90x | 24 | 74 | 1.64 | 1.98x |
| one | 256 | 104 | 73 | 8.25 | 13.26x | 58 | 42 | 3.87 | 8.08x |
| two | 256 | 46 | 154 | 3.73 | 2.83x | 21 | 87 | 1.54 | 1.54x |
| three | 256 | 43 | 169 | 3.58 | 2.52x | 52 | 169 | 3.55 | 1.93x |
| four | 256 | 56 | 126 | 4.59 | 4.31x | 61 | 115 | 3.90 | 3.22x |
| five | 256 | 43 | 155 | 3.48 | 2.70x | 49 | 155 | 3.20 | 1.90x |
| six | 256 | 48 | 150 | 3.93 | 3.13x | 20 | 90 | 1.47 | 1.40x |
| seven | 256 | 57 | 147 | 4.64 | 3.37x | 23 | 89 | 1.65 | 1.62x |
| eight | 256 | 48 | 155 | 3.82 | 2.84x | 57 | 154 | 3.83 | 2.28x |
| nine | 256 | 56 | 168 | 4.46 | 3.15x | 45 | 141 | 2.90 | 1.94x |
| 54.wcsp | 67 | 107.33 | 160.33 | 6.56 | 2.78x | 68.33 | 172 | 1.80 | 0.72x |
| or-chain-42 | 385 | 120.33 | 27 | 0.12 | 31.27x | 110 | 54.33 | 0.06 | 6.43x |
| or-chain-45 | 715 | 151 | 33.67 | 0.14 | 12.52x | 94.33 | 36.33 | 0.06 | 4.85x |
| or-chain-147 | 410 | 107.67 | 18.67 | 3.27 | 80.72x | 105 | 58.33 | 1.63 | 12.77x |
| or-chain-148 | 463 | 122.67 | 42.33 | 1.00 | 49.04x | 80 | 32 | 0.28 | 14.24x |
| or-chain-225 | 467 | 181.33 | 58 | 0.79 | 44.14x | 137.67 | 69 | 0.33 | 10.76x |
| rbm20 | 40 | 9 | 41 | 30.98 | 2.38x | 30 | 107.22 | 30.18 | 0.99x |
| Seg2-17 | 228 | 63 | 83.66 | 1.77 | 7.00x | 46.67 | 64.67 | 0.74 | 4.14x |
| Seg7-11 | 235 | 54.3 | 84 | 1.86 | 2.84x | 48.66 | 73.33 | 1.27 | 2.32x |
| Family2Dominant.1.5loci | 385 | 117.33 | 88 | 2.39 | 5.90x | 85.67 | 78.33 | 1.04 | 2.69x |
| Family2Recessive.15.5loci | 385 | 111.6 | 89.7 | 1.31 | 3.85x | 86.33 | 81.67 | 0.74 | 2.18x |
| grid10x10.f5.wrap | 100 | 136.67 | 239 | 17.36 | 6.26x | 142 | 180.33 | 10.30 | 4.63x |
| grid10x10.f10.wrap | 100 | 101.33 | 62.33 | 12.39 | 20.92x | 92.67 | 59 | 5.94 | 9.70x |
| **average** | **275.65** | **83.89** | **101.29** | **5.39** | **13.55x** | **66.84** | **94.89** | **3.56** | **4.45x** |

$\mathcal{D}\#(a,b,c) = 4$, $\mathcal{D}\#(a,b,\bar{c}) = 18$, $\mathcal{D}\#(a,\bar{b},c) = 2$, $\mathcal{D}\#(a,\bar{b},\bar{c}) = 13$, $\mathcal{D}\#(\bar{a},b,c) = 1$, $\mathcal{D}\#(\bar{a},b,\bar{c}) = 1$, $\mathcal{D}\#(\bar{a},\bar{b},c) = 42$, and $\mathcal{D}\#(\bar{a},\bar{b},\bar{c}) = 19$. Suppose we want to learn, from this dataset, a Markov network with 3 edges, $(A,B)$, $(B,C)$ and $(A,C)$, with the corresponding parameter sets $\theta_{AB}$, $\theta_{BC}$ and $\theta_{AC}$. If the initial set of parameters $\theta^\star = (\theta^\star_{AB}, \theta^\star_{BC}, \theta^\star_{AC})$ is uniform, i.e., $\theta^\star_{XY} = (1,1,1,1)$, then Equation 8 gives the sub-function $f_{\theta^\star}(\theta_{AB}) = -22 \cdot \log \theta_{ab} - 15 \cdot \log \theta_{a\bar{b}} - 2 \cdot \log \theta_{\bar{a}b} - 61 \cdot \log \theta_{\bar{a}\bar{b}}$. Moreover, we have $Z_{\theta^\star} = 2 \cdot \theta_{ab} + 2 \cdot \theta_{a\bar{b}} + 2 \cdot \theta_{\bar{a}b} + 2 \cdot \theta_{\bar{a}\bar{b}}$. Minimizing $f_{\theta^\star}(\theta_{AB})$ under $Z_{\theta^\star} = \alpha = 2$ corresponds to solving a convex optimization problem, which has the unique solution: $(\theta_{ab}, \theta_{a\bar{b}}, \theta_{\bar{a}b}, \theta_{a\bar{b}}) = (\frac{22}{100}, \frac{15}{100}, \frac{2}{100}, \frac{61}{100})$. We solve similar convex optimization problems for the other parameter sets $\theta_{BC}$ and $\theta_{AC}$, to update estimates $\theta^\star$. We then apply an appropriate feasibility function $g$ (see Footnote 7), and repeat until convergence.

## 6  Experimental Results

We evaluate now the efficiency of EDML, conjugate gradient (CG) and Limited-memory BFGS (L-BFGS), when learning Markov networks under complete data.[9] We first learned grid-structured pairwise MRFs from the CEDAR dataset of handwritten digits, which has 10 datasets (one for each digit) of $16\times16$ binary images. We also simulated datasets from networks used in the probabilistic inference evaluations of UAI-2008, 2010 and 2012, that are amenable to jointree inference.[10] For each network, we simulated 3 datasets of size $2^{10}$ examples each, and learned parameters using the original structure. Experiments were run on a 3.6GHz Intel i5 CPU with access to 8GB RAM.

We used the CG implementation in the Apache Commons Math library, and the L-BFGS implementation in Mallet.[11] Both are Java libraries, and our implementation of EDML is also in Java. More importantly, all of the CG, L-BFGS, and EDML methods rely on the same underlying engine for

exact inference.[12] For EDML, we damped parameter estimates at each iteration, which is typical for algorithms like loopy belief propagation, which EDML was originally inspired by [5].[13] We used Brent's method with default settings for line search in CG, which was the most efficient over all univariate solvers in Apache's library, which we evaluated in initial experiments.

We first run CG until convergence (or after exceeding 30 minutes) to obtain parameter estimates of some quality $q_{cg}$ (in log likelihood), recording the number of iterations $i_{cg}$ and time $t_{cg}$ required in minutes. EDML is then run next until it obtains an estimate of the same quality $q_{cg}$, or better, recording also the number of iterations $i_{edml}$ and time $t_{edml}$ in minutes. The time speed-up $S$ of EDML over CG is computed as $t_{cg}/t_{edml}$. We also performed the same comparison with L-BFGS instead of CG, recording the corresponding number of iterations $(i_{l\text{-bfgs}}, i'_{edml})$ and time taken $(t_{l\text{-bfgs}}, t'_{edml})$, giving us the speed-up of EDML over L-BFGS as $S' = t_{l\text{-bfgs}}/t'_{edml}$.

Table 1 shows results for both sets of experiments. It shows the number of variables in each network (#vars), the average number of iterations taken by each algorithm, and the average speed-up achieved by EDML over CG (L-BFGS).[14] On the given benchmarks, we see that on average EDML was roughly $13.5\times$ faster than CG, and $4.5\times$ faster than L-BFGS. EDML was up to an order-of-magnitude faster than L-BFGS in some cases. In many cases, EDML required more iterations but was still faster in time. This is due in part by the number of times inference is invoked by CG and L-BFGS (in line search), whereas EDML only needs to invoke inference once per iteration.

## 7    Related Work

As an iterative fixed-point algorithm, we can view EDML as a Jacobi-type method, where updates are performed in parallel [1]. Alternatively, a version of EDML using Gauss-Seidel iterations would update each parameter set in sequence using the most recently computed update. This leads to an algorithm that monotonically improves the log likelihood at each update. In this case, we obtain a coordinate descent algorithm, Iterative Proportional Fitting (IPF) [9], as a special case of EDML.

The notion of fixing all parameters, except for one, has been exploited before for the purposes of optimizing the log likelihood of a Markov network, as a heuristic for structure learning [15]. This notion also underlies the IPF algorithm; see, e.g., [13], Section 19.5.7. In the case of complete data, the resulting sub-function is convex, yet for incomplete data, it is not necessarily convex.

Optimization methods such as conjugate gradient, and L-BFGS [12], are more commonly used to optimize the parameters of a Markov network. For relational Markov networks or Markov networks that otherwise assume a feature-based representation [8], evaluating the likelihood is typically intractable, in which case one typically optimizes instead the pseudo-log-likelihood [2]. For more on parameter estimation in Markov networks, see [10, 13].

## 8    Conclusion

In this paper, we provided an abstract and simple view of the EDML algorithm, originally proposed for parameter estimation in Bayesian networks, as a particular method for continuous optimization. One consequence of this view is that it is immediate that fixed-points of EDML are stationary points of the log-likelihood, and vice-versa [16]. A more interesting consequence, is that it allows us to propose an EDML algorithm for a new class of models, Markov networks. Empirically, we find that EDML can more efficiently learn parameter estimates for Markov networks under complete data, compared to conjugate gradient and L-BFGS, sometimes by an order-of-magnitude. The empirical evaluation of EDML for Markov networks under incomplete data is left for future work.

**Acknowledgments**

This work has been partially supported by ONR grant #N00014-12-1-0423.

## Footnotes

[1]Under constraints, we consider points that are stationary with respect to the corresponding Lagrangian.

[2]We discuss this point further in the supplementary appendix.

[3]Theorem 1 assumes tacitly that $\theta_{x|\mathbf{u}}^\star \neq 0$. More generally, however, $C_{x|\mathbf{u}}^i = \partial Pr_{\theta^\star}(\mathbf{d}_i)/\partial \theta_{x|\mathbf{u}}$, which can also be computed using some standard inference algorithms [6, 14].

[4]Soft evidence is an observation that increases or decreases ones belief in an event, but not necessarily to the point of certainty. For more on soft evidence, see [4].

[5]Typically, we assume Dirichlet priors for MAP estimation. However, we focus on ML estimation here.

[6]More specifically, strict convexity implies a unique optimum, although under certain assumptions, we can guarantee that Equation 2 is indeed strictly convex.

[7]Here, $g(\theta_{\mathbf{x}_a})$ denotes the component of $g(\theta)$ corresponding to $\theta_{\mathbf{x}_a}$. Moreover, the function $g(\theta)$ can be constructed, e.g., by simply multiplying all entries of one parameter set by $\alpha/Z_\theta$. In our experiments, we

[9]We also considered Iterative Proportional Fitting (IPF) as a baseline. However, IPF does not scale to our benchmarks, as it invokes inference many times more often than the methods we considered.

[10]Network 54.wcsp is a weighted CSP problem; or-chain-{42, 45, 147, 148, 225} are from the Promedas suite; rbm-20 is a restricted Boltzmann machine; Seg2-17, Seg7-11 are from the Segmentation suite; family2-dominant.1.5loci, family2-recessive.15.5loci are genetic linkage analysis networks; and grid10x10.f5.wrap, grid10x10.10.wrap are 10x10 grid networks.

[11]Available at http://commons.apache.org/ and http://mallet.cs.umass.edu/.

[12] For exact inference in Markov networks, we employed a jointree algorithm from the SAMIAM inference library, http://reasoning.cs.ucla.edu/samiam/.

[13] We start with an initial factor of $\frac{1}{2}$, which we tighten as we iterate.

[14] For CG, we used a threshold based on relative change in the likelihood at $10^{-4}$. We used Mallet's default convergence threshold for L-BFGS.

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
