[Supplementary Material]

# EDML for Learning Parameters in Directed and Undirected Graphical Models

**Khaled S. Refaat, Arthur Choi, Adnan Darwiche**
Computer Science Department
University of California, Los Angeles
{krefaat,aychoi,darwiche}@cs.ucla.edu

## A   Proofs

**Proof of Theorem 1**   First, the probability of an example $\mathbf{d}_i \in \mathcal{D}$ is:

$$Pr_\theta(\mathbf{d}_i) = \sum_{\mathbf{x} \sim \mathbf{d}_i} \prod_{x|\mathbf{u} \sim \mathbf{x}} \theta_{x|\mathbf{u}}$$

where operator $\sim$ denotes compatibility between two instantiations (they set the same value to common variables). For a fixed parameter set $\theta_{X|\mathbf{u}}$, the probability $Pr_\theta(\mathbf{d}_i)$ is a linear function with respect to the parameters of $\theta_{X|\mathbf{u}}$:

$$Pr_\theta(\mathbf{d}_i) = Pr_\theta(\neg\mathbf{u}, \mathbf{d}_i) + \sum_x Pr_\theta(x\mathbf{u}, \mathbf{d}_i)$$

$$= Pr_\theta(\neg\mathbf{u}, \mathbf{d}_i) + \sum_x \frac{\partial Pr_\theta(\mathbf{d}_i)}{\partial \theta_{x|\mathbf{u}}} \theta_{x|\mathbf{u}}$$

$$= C_\mathbf{u}^i[\theta] + \sum_x C_{x|\mathbf{u}}^i[\theta] \cdot \theta_{x|\mathbf{u}}$$

where $C_\mathbf{u}^i[\theta]$ and $C_{x|\mathbf{u}}^i[\theta]$ are constants with respect to $\theta_{X|\mathbf{u}}$. Moreover $Pr_\theta(\neg\mathbf{u}, \mathbf{d}_i) = Pr_\theta(\mathbf{d}_i) - Pr_\theta(\mathbf{u}, \mathbf{d}_i)$. Thus our sub-function, the negative log-likelihood with respect to parameter set $\theta_{X|\mathbf{u}}$, has the form:

$$f_{\theta^\star}(\theta_{X|\mathbf{u}}) = -\sum_{i=1}^N \log\left( C_\mathbf{u}^i[\theta^\star] + \sum_x C_{x|\mathbf{u}}^i[\theta^\star] \cdot \theta_{x|\mathbf{u}} \right).$$

$\square$

**Proof of Theorem 2**   The log-likelihood of soft evidence in this model is:

$$\log \mathbb{P}(\eta|\theta_{X|\mathbf{u}}) = \sum_{i=1}^N \log \mathbb{P}(\eta_i|\theta_{X|\mathbf{u}})$$

$$= \sum_{i=1}^N \log \sum_{x_i} \mathbb{P}(\eta_i|x_i, \theta_{X|\mathbf{u}}) \mathbb{P}(x_i|\theta_{X|\mathbf{u}})$$

$$= \sum_{i=1}^N \log \sum_{x_i} \mathbb{P}(\eta_i|x_i) \cdot \theta_{x|\mathbf{u}}.$$

If we substitute $\mathbb{P}(\eta_i|x_i) = C_\mathbf{u}^i[\theta^\star] + C_{x|\mathbf{u}}^i[\theta^\star]$, we have

$$\log \mathbb{P}(\eta|\theta_{X|\mathbf{u}}) = \sum_{i=1}^N \log \sum_x \left( C_\mathbf{u}^i[\theta^\star] + C_{x|\mathbf{u}}^i[\theta^\star] \right) \cdot \theta_{x|\mathbf{u}}$$

$$= \sum_{i=1}^N \log \left( C_\mathbf{u}^i[\theta^\star] + \sum_x C_{x|\mathbf{u}}^i[\theta^\star]\theta_{x|\mathbf{u}} \right)$$

which is Equation 2, negated. $\square$

**Proof of Theorem 3**   Suppose that $\theta^*$ is optimal for Equation 6. Multiplying an arbitrary $\theta_{\mathbf{X}_a}^*$ by a constant, results in multiplying both $Z_\theta$, and $Z_\theta(\mathbf{d}_i)$, by the same constant, which cancels out in each pair of terms, $\log Z_\theta - \log Z_\theta(\mathbf{d}_i)$, preserving the same optimal objective value. Thus, one could always find an optimal $\theta$ where $\theta_{\mathbf{X}_a} \propto \theta_{\mathbf{X}_a}^*$, that is optimal for Equation 6, and where $Z_\theta = \alpha$.

Thus, fixing $Z_\theta = \alpha$ does not exclude the optimal solution for Equation 6, which can be now reduced to:

$$f(\theta) = N \log \alpha - \sum_{i=1}^N \log Z_\theta(\mathbf{d}_i) \tag{9}$$

with a feasibility constraint that $Z_\theta = \alpha$.

Equation 9 is equivalent to Equation 7, since $N \log \alpha$ is a constant. As a result, if $g(\theta)$ is feasible and optimal for Equation 7, then any $\theta$, where $\theta_{\mathbf{X}_a} \propto g(\theta_{\mathbf{X}_a}) \, \forall \, \mathbf{X}_a$ is optimal for Equation 6. We will next prove the second part of the theorem.

The partial derivative of the log likelihood $\ell\ell(\theta|\mathcal{D})$ w.r.t. parameter $\theta_{\mathbf{x}_a}$ is:

$$\frac{\partial \ell\ell}{\partial \theta_{\mathbf{x}_a}} = -\frac{N}{Z_\theta} \frac{\partial Z_\theta}{\partial \theta_{\mathbf{x}_a}} + \sum_{i=1}^{N} \frac{1}{Z_\theta(\mathbf{d}_i)} \frac{\partial Z_\theta(\mathbf{d}_i)}{\partial \theta_{\mathbf{x}_a}}.$$

First, note that:

$$\frac{1}{Z_\theta} \frac{\partial Z_\theta}{\partial \theta_{\mathbf{x}_a}} \theta_{\mathbf{x}_a} = Pr(\mathbf{x}_a), \quad \frac{1}{Z_\theta(\mathbf{d}_i)} \frac{\partial Z_\theta(\mathbf{d}_i)}{\partial \theta_{\mathbf{x}_a}} \theta_{\mathbf{x}_a} = Pr_\theta(\mathbf{x}_a | \mathbf{d}_i)$$

Thus, with some re-arranging, we obtain:

$$Pr_\theta(\mathbf{x}_a) = \frac{1}{N} \sum_{i=1}^{N} Pr_\theta(\mathbf{x}_a | \mathbf{d}_i) \tag{10}$$

which is the "moment matching" condition for parameter estimation in Markov networks. Second, consider the simplified objective: $f(\theta) = -\sum_{i=1}^{N} \log Z_\theta(\mathbf{d}_i)$ which is subject to the constraint $Z = \alpha$. We construct the Lagrangian $L(\theta, \nu) = f(\theta) + \nu(Z - \alpha)$. Setting to zero the partial derivative w.r.t. $\nu$, we obtain our constraint $Z = \alpha$. The partial derivative w.r.t. parameter $\theta_{\mathbf{x}_a}$ is:

$$-\sum_{i=1}^{N} \frac{1}{Z_\theta(\mathbf{d}_i)} \frac{\partial Z_\theta(\mathbf{d}_i)}{\partial \theta_{\mathbf{x}_a}} + \nu \frac{\partial Z_\theta}{\partial \theta_{\mathbf{x}_a}}.$$

We set the partial derivative to zero, multiply the second term by $\frac{\alpha}{Z} = 1$, and re-arrange, giving us:

$$\nu \alpha Pr_\theta(\mathbf{x}_a) = \sum_{i=1}^{N} Pr_\theta(\mathbf{x}_a \mid \mathbf{d}_i).$$

Summing each equation for all instantiations $\mathbf{x}_a$, we identify $\nu = \frac{N}{\alpha}$, which after substitution, gives us a condition equivalent to Equation 10.

Note that the stationary condition given by Equation 10 depends only on marginals, not the absolute value of the partition function. Moreover, applying a proper feasibility function $g(\theta)$, where $\theta_{\mathbf{X}_a} \propto g(\theta_{\mathbf{X}_a}) \, \forall \, \mathbf{X}_a$, will not change the marginals implied by $\theta$, as the multiplicative factors cancel out in each pair of terms, $\log Z_\theta - \log Z_\theta(\mathbf{d}_i)$. Thus if a point $\theta$ satisfies Equation 10, then $g(\theta)$ must also satisfy it. Similarly, if $g(\theta)$ satisfies Equation 10, the original point $\theta$ must also satisfy it. $\quad\square$

**Proof of Theorem 4** First, the partition function conditioned on an example $\mathbf{d}_i \in \mathcal{D}$ is:

$$Z_\theta(\mathbf{d}_i) = \sum_{\mathbf{x} \sim \mathbf{d}_i} \prod_{\mathbf{x}_a \sim \mathbf{x}} \theta_{\mathbf{x}_a}$$

where operator $\sim$ denotes compatibility between two instantiations (they set the same value to common variables). For a given parameter set $\theta_{\mathbf{X}_a}$, the partition function $Z_\theta(\mathbf{d}_i)$ is a linear function with respect to the parameters $\theta_{\mathbf{X}_a}$:

$$Z_\theta(\mathbf{d}_i) = \sum_{\mathbf{x}_a} Z_\theta(\mathbf{x}_a, \mathbf{d}_i) = \sum_{\mathbf{x}_a} \frac{\partial Z_\theta(\mathbf{d}_i)}{\partial \theta_{\mathbf{x}_a}} \theta_{\mathbf{x}_a}$$

$$= \sum_{\mathbf{x}_a} C^i_{\mathbf{x}_a}[\theta] \cdot \theta_{\mathbf{x}_a}$$

where $C^i_{\mathbf{x}_a}[\theta]$ is a constant with respect to $\theta_{\mathbf{X}_a}$. Thus, our sub-function, has the form:

$$f_{\theta^\star}(\theta_{\mathbf{X}_a}) = -\sum_{i=1}^{N} \log \sum_{\mathbf{x}_a} C^i_{\mathbf{x}_a}[\theta^\star] \cdot \theta_{\mathbf{x}_a}.$$

Figure 2: A meta network induced from a base network $S \longleftarrow H \longrightarrow E$. The CPTs here are based on standard semantics; see, e.g., [8], Ch. 18.

On the other hand, the constraint $Z_\theta = \alpha$ takes the form:

$$Z_\theta = \sum_{\mathbf{x}_a} Z_\theta(\mathbf{x}_a) = \sum_{\mathbf{x}_a} \frac{\partial Z_\theta}{\partial \theta_{\mathbf{x}_a}} \theta_{\mathbf{x}_a} = \sum_{\mathbf{x}_a} C_{\mathbf{x}_a} \theta_{\mathbf{x}_a} = \alpha$$

$\square$

**Theorem 5** *Suppose we have a feasibility function*

$$g(y_1, \ldots, y_n) = (x_1, \ldots, x_n)$$

*where $x_i \neq y_i$ implies that the point $(x_1, \ldots, y_i, \ldots, x_n)$ is infeasible (e.g., Euclidean projection satisfies this condition). Suppose now that the algorithm produces a sequence $x^t, y^{t+1}, x^{t+1} = x^t$. Then $x^t$ must be a feasible and stationary point.*

**Proof** By the statement of the iterative procedure, $x^t$ is guaranteed to be feasible. Suppose that $g(y^{t+1}) = x^{t+1} = x^t$. First, it must be that $y^{t+1} = x^t$. Suppose instead that $y^{t+1} \neq x^t$, and thus for some component, $y_i^{t+1} \neq x_i^t$. By our feasibility function, $(x_1^t, \ldots, y_i^{t+1}, \ldots, x_n^t)$ must be infeasible. However, Step 2(a) ensures that $(x_1^t, \ldots, y_i^{t+1}, \ldots, x_n^t)$ is feasible. Hence, it must be that $y^{t+1} = x^t$. Further, by Step 2(a) and Claim 1, $x^t$ must also be stationary. $\square$

# B  A Review of EDML

EDML is a recent method for learning Bayesian network parameters from incomplete data [6, 18]. It is based on Bayesian learning in which one formulates estimation in terms of computing posterior distributions on network parameters. That is, given a Bayesian network, one constructs a corresponding *meta network* in which parameters are explicated as variables, and on which the given dataset $\mathcal{D}$ can be asserted as evidence; see Figure 2. One then estimates parameters by considering the posterior distribution obtained from conditioning the meta network on the given dataset $\mathcal{D}$. Suppose for example that the meta network induces distribution $\mathcal{P}$ and let $\theta$ denote an instantiation of variables that represent parameters in the meta network. One can then obtain MAP parameter estimates by computing $\text{argmax}_\theta \mathcal{P}(\theta|\mathcal{D})$ using inference on the meta network.

It is known that meta networks tend to be too complex for exact inference algorithms, especially when the dataset is large enough. The basic insight behind EDML was to adapt a specific approximate inference scheme to meta networks with the goal of computing MAP parameter estimates. In particular, the original derivation of EDML adapted the approximate inference algorithm proposed by [5], in which edges are deleted from a Bayesian network to make it sparse enough for exact inference, followed by a compensation scheme that attempts to improve the quality of the approximations obtained from the edge-deleted network. The adaptation of this inference method to meta networks is shown in Figure 3. The two specific techniques employed here were to augment each edge $\theta_{X|\mathbf{u}} \longrightarrow X^i$ by an auxiliary variable $X_{\mathbf{u}}^i$, leading to $\theta_{X|\mathbf{u}} \longrightarrow X_{\mathbf{u}}^i \longrightarrow X^i$, where $X_{\mathbf{u}}^i \longrightarrow X^i$ is

Figure 3: An edge-deleted network obtained from the meta network in Figure 2. Highlighted are the island for example $\mathbf{d}_2$ and the island for parameter set $\theta_{S|h}$.

---

**Algorithm 1** EDML

---

**input:**

| | |
|---|---|
| $G$: | A Bayesian network structure |
| $\mathcal{D}$: | An incomplete dataset $\mathbf{d}_1, \ldots, \mathbf{d}_N$ |
| $\theta$: | An initial parameterization of structure $G$ |
| $\alpha_{X|\mathbf{u}}, \beta_{X|\mathbf{u}}$: | Beta prior for each random variable $X|\mathbf{u}$ |

1: **while** not converged **do**
2:     $Pr \leftarrow$ distribution induced by $\theta$ and $G$
3:     **Compute** Bayes factors:

$$\kappa_{x|\mathbf{u}}^i \leftarrow \frac{Pr(x\mathbf{u}|\mathbf{d}_i)/Pr(x|\mathbf{u}) - Pr(\mathbf{u}|\mathbf{d}_i) + 1}{Pr(\bar{x}\mathbf{u}|\mathbf{d}_i)/Pr(\bar{x}|\mathbf{u}) - Pr(\mathbf{u}|\mathbf{d}_i) + 1}$$

    for each family instantiation $x\mathbf{u}$ and example $\mathbf{d}_i$
4:     **Update** parameters:

$$\theta_{x|\mathbf{u}} \leftarrow \underset{p}{\operatorname{argmax}} \; [p]^{\alpha_{X|\mathbf{u}}-1}[1-p]^{\beta_{X|\mathbf{u}}-1} \prod_{i=1}^{N} [\kappa_{x|\mathbf{u}}^i \cdot p - p + 1]$$

5: **return** parameterization $\theta$

---

an equivalence edge. This is followed by deleting the equivalence edge. This technique yielded a disconnected meta network with two classes of sub-networks, called *parameter islands* and *network islands.*

Deleting edges, as proposed by [5], leads to introducing two auxiliary nodes in the Bayesian network for each deleted edge. Moreover, approximate inference by edge deletion follows the deletion process by a compensation scheme that searches for appropriate CPTs of these auxiliary nodes. As it turns out, the search for these CPTs, which is done iteratively, was amenable to a very intuitive interpretation as shown in [6].

In particular, one set of CPTs corresponded to soft evidence on network parameters, where each network island contributes one piece of soft evidence for each network parameter. The second set of CPTs corresponded to updated parameter estimates, where each parameter island contributes an estimate of its underlying parameter set. This interpretation was the basis for the form of EDML shown in Algorithm 1.[15] EDML iterates just like EM does, producing new estimates after each

iteration. However, EDML iterations can be viewed as having two phases. In the first phase, each example in the data set is used to compute a piece of soft evidence on each parameter set (Line 3 of Algorithm 1). In the second phase, the pieces of soft evidence pertaining to each parameter set are used to compute a new estimate of that set (by solving the convex optimization problem on Line 4 of Algorithm 1). The process repeats until some convergence criteria is met. Aside from this optimization task, EM and EDML have the same computational complexity.

## Footnotes

[15]This form is specific to binary variables; a multi-valued generalization was provided in [18].