[Reviews · NeurIPS 2013]

Submitted by Assigned_Reviewer_6

The authors claim to represent a simplified perspective of EDML, that:
1. casts it as a general approach to continuous optimization
2. makes immediate some results that were previously obtained for EDML, but through more effort
3. Facilitates the design of new EDML algorithms for new classes of models

First, EDML in Bayesian networks is reviewed as a method to optimize the likelihood of the observed data, where each update step for the next parameters requires inference on the Bayesian network using the current parameters. Then, the same analysis is applied to undirected models (MRFs). As a last step, EDML is claimed to converge in one iteration for MRFs under complete data. The last is experimented and compared to conjugate gradient (CG) and L-BFGS on benchmark problems, including handwritten digits modeled with 2D-grid-MRFs and 13 other networks that could be infered with jointree. EDML outperforms both CG and L-BFGS in running time till convergence to a certain accuracy, even if number of iterations may be larger at times.

Comments:
- The title is very similar to UAI10 and does not emphasize the main point claimed for this particular work, which is "observing EDML as a general approach for continuous optimization under complete data."
- Why would the overview of EDML in the uai paper not fall under the definition of "continuous optimization"? Not sure I see the difference in "perspective"
- To my understanding, the focus in previous work was application to Bayesian networks and comparison to EM, while here the focus is undirected MRFs and comparison to CG and L-BFGS. The experiments indeed show the benefit in using this method upon the other optimization techniques.
Summary: The main contribution is providing a useful technique for parameter training in MRFs with complete data, by "transferring" EDML from Bayesian (directed) to undirected networks.
It wasn't clear to me why this is claimed to be a different view of the method.

Submitted by Assigned_Reviewer_7

The paper provides a simplified perspective on EDML
and illustrated the benefits of such a simplified view.
First, existing results about EDML are much easier to proce.
Second, and most important, this new formulation provides
a systematic procedure for deriving new instances of EDML
for other models. This is illustrated for Markov networks.
The derived EDML is shown to be competitive to existing
learning approaches.

Parameter estimation of graphical models is important.
The paper, which is extremely well written,
presents a very elegant view on this task.
I also like the simplified view on EDML. It really helps
understanding it pros and cons. It also shows that
it essentially implements a coordinate descend for
parameter estimation. Indeed, such approach have been
proposed for graphical models --- for instance,
iterative proportional fitting can be viewed as such a
method --- however I expect it to be too slow
to scale well. Still, there are some recent advances
for MaxEnt models

Fang-Lan Huang, Cho-Jui Hsieh, Kai-Wei Chang, Chih-Jen Lin:
Iterative Scaling and Coordinate Descent Methods for Maximum Entropy Models.
Journal of Machine Learning Research 11: 815-848 (2010)

that is also nicely comparing iterative scaling and coordinate
descend. Moreover, it shows that there is not just one IPS
approach in general but several. It just depends on the
auxiliar function you are using. This should really be clarified.

In my opinion, the big pro of the present paper is that it shows that
parameter estimation within graphical models requires to
iteratively solve rather simple convex optimization sub-problems.
In many cases, each sub-problem is of simple form, too, and
involved inferences made from the data at hand. I value this
clear view on what parameter estimation is a lot.

However, there are also other gradient optimization techniques
that avoid the line-search and have an interesting connection
to EM such as

E. Bauer, D. Koller, Y. Singer:
Update Rules for Parameter Estimation in Bayesian Networks.
UAI 1997: 3-13

L.E. Ortiz, L. Pack Kaelbling:
Accelerating EM: An Empirical Study.
UAI 1999: 512-521

J. Fischer, K. Kersting:
Scaled CGEM: A Fast Accelerated EM.
ECML 2003: 133-144

M. Jamshidian and R. I. Jennrich.
Accleration of the EM Algorithm by using Quasi-Newton Methods.
Journal of the Royal Stat. Society B, 59(3):569–587, 1997.

I would expect these approaches to work almost as good but to run
considerable faster than CG and LBFGS.

Still the transfer to the Markov network learning setting is
interesting.

Summary: To summarize,

+ Novel perspective on EDML that provides additional insight into it.
+ Derives an EDML approach for parameter learning of Markov networks

- connection to CS and IPS approaches should be improved and also
illustrated empirically
- Many accelerated parameter estimation method are not discussed.
For instance, there are accelerated EM approaches that also make
use of advanced gradient methods such as scaled-conjugate gradient
avoiding the line-search via scaling an (approx.) Hessian.

Although I still think that the novel view is highly interesting, the discussion also
revealed that some issues (see the other reviews) should be addressed. Still, since I think the simplified vies is interesting I lean towards accept.

Submitted by Assigned_Reviewer_8

Summary:
Overall, I think the paper is potentially useful and a contribution to the community since it puts forth a simpler framework for EDML (the previous works on EDML are quite dense). However, I was disappointed that it presented it in a confusing manner and that it didn't address a number of key issues such as convergence, relationship to EM, etc. Further, the contribution to parameter estimation for undirected models is quite incremental.

Contribution:
This paper portrays EDML as fixed-point iteration where fixed points correspond to stationary points of the log likelihood. Each iteration requires performing inference and has a notion of soft evidence, like EM. Their application to complete-data learning for Markov networks reduces to some Jacobi-style/ coordinate descent-type method for the convex log likelihood and can be used as a black-box optimizer for Markov networks.

Novelty:
The connection between EDML and continuous optimization is novel. However, the literature on alternatives to EM is vast, and I'm not sure how novel it is as a learning method under missing data. For complete-data Markov networks, the method is similar to basic, known methods for convex optimization.

Pros:
Attractive method for learning in the presence of missing data that points to fertile ground for future work. Unified treatment of directed and undirected graphical models is also interesting and potentially impactful. Many practitioners would benefit from reading the paper in order to think about alternative perspectives on parameter learning in the presence of unobserved data.

Cons:
1) The paper casts EDML as an optimization technique, but lacks necessary details for readers to understand this interpretation.
2) Exposition, particularly notation, is sometimes confusing.
3) insufficient conceptual or experimental comparison to EM.
4) experiments only address an overly simple case of parameter learning problems (complete data).

1) I found section 2 very confusing. It first appears as though you're introducing some new family of optimization algorithms, but then it becomes apparent that you're discussing the general family of techniques described in the first paragraph of the related work section. I would reshape this section to first state that such a family exists and discuss their convergence properties. Then, I'd state that for the particular class of problems you're dealing with, these coordinate-wise minimizations are tractable. Claim1 is boring, and more space should be made for previewing the sort of content discussed in the first paragraph of the related work section.

The paper also lacks a formal discussion of convergence behavior for the method. How do we know that the update rule at the top of page 5 will converge? I agree that its fixed points are fixed points of the marginal likelihood, but how do we know it will ever converge to one of these? This issue would be remedied if you were much more straightforward in section 2 about what you need for such coordinate minimization techniques to converge. Fixed point iteration is not guaranteed to converge, you need some sort of contraction. See proofs, for example, about the convergence of loopy belief propagation.

Why does section 3.4 have the title 'Finding the Unique Optimum?" The subfunctions are convex, so each of them has a unique minimizer. However, the overall problem doesn't have a unique stationary point, and the algorithm you present in this section is for finding such a point. I understand that the updates at the top of page 5 are obtained by analytically minimizing simultaneously in every coordinate (where you find a unique minimum), but the title is very confusing.

2) The notation in section 3.2 (soft evidence) needs to be defined much more clearly. In equation (3) is there implicitly a sum over possible values that X_i can take on? I don't understand how you can get a marginal likelihood without summing over the possible values for each X_i. What's the difference between P(\eta_i|x) and P(X_i|x)? The corresponding appendix does not clear things up either. Furthermore, I didn't find 3.2 particularly illuminating about the method without some comparison to 'soft evidence' in EM.

Also, it's strange that you present an algorithm at the top of page 5 based on an earlier work and leave it to the reader to deduce that this can also be obtained by applying the procedure from page 2 to the log likelihood. You should, at the very least, say that the updates at the top of page 5 correspond to exact simultaneous minimization of the subfunctions. You should also discuss under what conditions such fixed point iteration will converge.

3) In general, I'd like to see much more discussion about the relationship between EDML and EM. EM has some desirable properties (monotonicity of the updates) and some downsides (it's maximizing a concave lower bound on the marginal likelihood rather than the actual likelihood). It would be good to juxtapose these with the properties of EDML. Of course it would have been nice to see experiments comparing the two as well.

4) The experiments section only considers complete-data learning, in which the log likelihood is convex. In this setting EDML is just another convex optimization method, so a much more rigorous discussion of its convergence behavior, memory requirements, per-iteration complexity, robustness to errors in approximate inference, etc. is necessary. For future work, I would make sure to do experiments with missing data and compare to EM. The two methods are not guaranteed to converge to the same value of the marginal likelihood, so comparing solution quality, and not just time, is necessary.

Minor points
What exactly is the semantics of your feasibility function g(y)? Are you always assuming it's a projection onto the feasible set? Is the feasibility function you use in If you don't, g() isn't guaranteed to give you what you want. First, you make it sound like a projection function onto a feasible set, but then you use it as an indicator function for the set?
You should define somewhere early on what EDML stands for.
Is this technique stable if doing exact inference in the bayes net is intractable? Discussion or experimental evaluation in this setting would be useful.
The update equation at the top of page 5 should have a number.

----- Post Author Response -----
In their response, the authors address many of the key omissions in their paper, such as the lack of a proof of convergence, by referring us to their other recent papers on EDML. This isn't quite sufficient to address the confusion that will arise for future readers of the paper. Some things, such as a comparison to EM can be addressed by providing a pointer to the earlier papers. However, the discussion of EDML as an optimization technique needs to be able to stand on its own. It is not sufficient to present an optimization algorithm and not prove that it converges or characterize what sorts of points it will converge to.

Overall, I am familiar with the modern literature on inference and learning in graphical models, and this paper is not of sufficient quality to be presented at NIPS. My concerns with this paper are mostly presentation related. For subsequent versions of the paper, I hope the authors will address the concerns that were raised. This is quite an important contribution, and I hope authors will resubmit it again (if it is rejected at NIPS).
Summary: I think the paper is potentially useful and a contribution to the community since it puts forth a simpler framework for EDML (the previous works on EDML are quite dense). However, I was disappointed that it presented it in a confusing manner and that it didn't address a number of key issues such as convergence, relationship to EM, etc. Further, the contribution to parameter estimation for undirected models is quite incremental.
Author Feedback

Author rebuttal: One goal of this work is to indeed provide a different perspective on parameter estimation in probabilistic graphical models (directed and undirected alike). Namely, we seek an understanding of parameter estimation beyond what was provided by classical and well-studied frameworks such as EM. Reviewer 7 clearly appreciates this view and the insights it brings. Reviewer 8 recognizes the fertile ground that this perspective may lay for future work. Reviewer 6 recognizes the utility of this view in deriving new EDML algorithms for undirected models.

Reviewer 8 asked for a comparison between EDML and EM. Please note that previous work [5, 16] has already compared EDML to EM, theoretically and empirically. Given the proposed pure optimization perspective, we thought it was more appropriate to use the limited space available to relate EDML to existing optimization methods (such as CG and L-BFGS), under complete data. Our experiments also imply practical benefits for the incomplete data case, when EDML is used in lieu of gradient methods for the M-step of EM (comparisons with EM were also given for directed models in [5, 16]). On the other hand, [5, 16] include answers to several questions that Reviewer 8 raised, including discussions about convergence. We will next clarify the main points raised by each reviewer.

Reviewer 6
**********

#1 (The UAI paper):

1a) EDML in the UAI paper is a "specific" optimization algorithm for learning Bayesian networks (BNs) that invokes a lot of BN concepts.

1b) Viewing EDML in the simple form given in Section 2 did not invoke any BN concepts, and therefore interpreting EDML as a "general" optimization algorithm.

#2 (Other comments):

2a) For MRFs, EDML can take more than one iteration to converge under complete data (Lines 301 and 302).

2b) The title emphasizes that EDML can be now used for both directed and undirected models. We will think about a more informative title.

Reviewer 7
**********

#1 (Parameter estimation in graphical models):

1a) The reviewer's note about the big pro of the paper is insightful: showing that parameter estimation in graphical models requires iteratively solving rather simple convex optimization sub-problems.

1b) We emphasize too that the creation of all these sub-problems requires performing inference "only once", at every EDML iteration, which we think is interesting.

#2 (EDML as a coordinate descent (CD) method):

2a) EDML as a CD method makes two choices:
1] Optimization in different directions is done in parallel, i.e. the current parameters are used to generate all sub-problems.
2] Every parameter set is considered a direction of optimization.

2b) Huang et al 2010, proposed a unified framework. They discuss CD methods that use sequential and parallel updates. EDML falls in the latter class. We will clarify the connection in the paper.

#3 (Other comments):

3a) The reviewer points to interesting related work that we will definitely consider. We tried simple gradient descent (with and "without" line search), but did not find it competitive on our benchmarks. There are still many variations that are worth trying, as suggested by the reviewer.

Reviewer 8
**********

#1 (EDML vs EM):

1a) A theoretical and empirical comparison between EDML and EM was already done in previous work [5, 16], and showed advantages for EDML. The goal of this paper is not to re-compare EDML with EM, but to simplify and generalize EDML.

#2 (Sec 3.4):

2a) This section does not find a stationary point for the overall learning problem, but only finds the unique optimum of each convex sub-problem. This is only a single step of the EDML algorithm. At the beginning of Section 3.4, we mention explicitly that we are solving Equation 3 (the sub-problem).

#3 (Convergence and the convex sub-problems):

3a) The convergence analysis of the update equation solving the small convex sub-problems was already given in [16]: the update equation at the top of Page 5 is a simpler form of (Equation 4 in [16]), which was proved to converge by (Theorem 2 in [16]).

3b) While (Equation 4 in [16]) is guaranteed to converge, the EDML iterative procedure is not guaranteed to converge in general cases. Please refer to (Section 6 in [5]) for cases where EDML is guaranteed to converge.

3c) Equation 2 is a simplified form of the convex sub-problems gotten by previous work [16], and therefore it is possible to use (Equation 4 in [16]) to optimize it, as proposed in Sec 3.4.

We will clarify these explicitly in the paper.

#4 (Markov Networks and Experiments):

4a) Beyond allowing us to derive EDML for Markov networks, our new formulation further lends itself to the derivation of EDML for new families of graphical models.

4b) In our experiments, the partition function is still non-trivial to compute. Thus, even with complete data, minimizing the number of evaluations of the partition function or its gradient is quite important.

#5 (Section 2 and the Related Work):

5a) Section 2 sheds light on a family of algorithms, emphasizing that the current feasible parameters are used to generate all the sub-problems. IPF does not fall directly in this family. We will clarify such difference in Section 2.

#6 (Soft Evidence):

6a) There is a specific technical definition for the term soft evidence we use (See Footnote 4). P(\eta_i|x) and P(X_i|x) are different; details are given in [4,16] but we will clarify in this paper as well.

6b) sum_x is a summation over all the possible values of X. Note that Equation 3 is for one sub-problem. Sub-problems become independent.

#7 (Other comments):

7a) In Section 2, the feasibility function (g) takes an arbitrary point and returns a feasible point. Theorem 5 in the appendix adds a constraint on g to get some favorable behavior. We do not use g as an indicator function.

7b) We make no claims yet about the algorithm's behavior with approximate inference.